# Surgery for Inflammatory Bowel Disease in the Era of Biologic Therapy: A Multicenter Experience from Romania

**DOI:** 10.3390/medicina59020337

**Published:** 2023-02-10

**Authors:** Christopher Pavel, Mircea Diculescu, Gabriel Constantinescu, Oana-Mihaela Plotogea, Vasile Sandru, Corina Meianu, Ion Dina, Ioana Pop, Andreea Butuc, Mariana Mihaila, Madalina Stan-Ilie

**Affiliations:** 1Department 5, Carol Davila University of Medicine and Pharmacy, 050474 Bucharest, Romania; 2Department of Gastroenterology, Clinical Emergency Hospital of Bucharest, 014461 Bucharest, Romania; 3Department of Gastroenterology, Fundeni Clinical Institute, 022328 Bucharest, Romania; 4Department of Gastroenterology, Sf. Ioan Clinical Emergency Hospital, 042122 Bucharest, Romania

**Keywords:** Crohn’s disease, ulcerative colitis, surgery, biologics, complications

## Abstract

*Background and Objectives:* Biologic therapy has fundamentally changed the opportunity of medical treatment to induce and maintain remission in inflammatory bowel disease (IBD). Nevertheless, the rate of surgery is still at a very high rate, profoundly affecting the quality of life. We aimed to analyze surgical cases at three major IBD units in order to identify the main risk factors and the impact of biologic therapy on pre- and postsurgical outcomes. *Material and Methods:* This was a multicenter retrospective cohort study that included 56 patients with IBD-related surgical interventions from 3 tertiary care hospitals in Bucharest, Romania. The study was conducted between January 2017 and June 2021. All data were retrospectively collected from the medical records of the patients and included the age at diagnosis, age at the time of surgery, IBD type and phenotype, biologic therapy before or/and after surgery, timing of biologic therapy initiation, extraintestinal manifestations, type of surgery (elective/emergency), early and long-term postoperative complications and a history of smoking. *Results:* A low rate of surgical interventions was noted in our cohort (10.3%), but half of these occurred in the first year after the IBD diagnosis. A total of 48% of the surgical interventions had been performed in an emergency setting, which seemed to be associated with a high rate of long-term postoperative complications. We found no statistically significant differences between IBD patients undergoing treatments with biologics before surgery and patients who did not receive biologics before the surgical intervention in terms of the IBD phenotype, type of surgery and postoperative complications. *Conclusion:* Our study showed that biologics initiated before the surgical intervention did not influence the postoperative complications. Moreover, we demonstrated that patients with Crohn’s disease and no biologics were the most susceptible to having to undergo surgery. Conclusion: In conclusion, the management of patients with IBD requires a multidisciplinary approach that considers an unpredictable evolution.

## 1. Introduction

Inflammatory bowel diseases (IBDs) are represented by idiopathic conditions characterized by chronic and dysregulated immune activation within the gastrointestinal tract in genetically susceptible individuals, with an accelerating incidence worldwide. Although biologic therapy has become a key component in the management of the disease since the approval of infliximab by the FDA in 1998, surgery is still required in almost half of patients at 10 years after the diagnosis and about one-third of patients require a second operation within 5 years after the first [1,2]. It is important to note that repeated surgical interventions are usually required in Crohn’s disease (CD) because the objective is to treat the complications (e.g., intractable fibrotic stricture) whereas in ulcerative colitis (UC), it is frequently a consequence of a medical therapy failure leading to fulminant colitis. As noted by Wong et al., there is conflicting data regarding the role of biologic therapy in reducing the postoperative recurrence and the need for secondary surgical interventions [3]; there are studies where biologic therapy improved endoscopic recurrence [4] whereas other studies did not show any superiority of biologics [5]. The surgical approach should not be changed in CD patients with a preoperative exposure to a biologic therapy whereas in UC, a 3-stage of a modified 2-stage ileal pouch–anal anastomosis (IPAA) to delay the pouch formation should be performed in order to prevent postoperative infectious complications [6].

There is a concern that the mucosal healing process induced by biologic therapy may lead to strictures or exacerbate existing obstructive lesions, leading to critical fibrotic strictures amenable only to endoscopic or surgical procedures [7]. One multicentric trial led by D’Haens et al. evaluated 30 patients from an endoscopic and a histologic point of view regarding mucosal healing after infliximab treatment; there was no remission in the pre-existing strictures and, additionally, 1 patient developed a new stricture at a site of severe ulceration [8]. Toy et al. followed 10 patients with stricturing CD, which showed a progression to a complete bowel obstruction that required a surgical intervention less than 2 months after infliximab initiation [9]. As observed by Bharadwaj et al., all major randomized control trials of biologic therapy (including infliximab, adalimumab, certolizumab, natalizumab, vedolizumab and ustekinumab) have excluded patients with previously known strictures; hence, interpreting their potential beneficial effects in this particular aspect (intestinal obstruction) is not feasible [10]. Furthermore, as inflammatory bowel diseases are not curative, the rate of recurrence after a surgical intervention is high (for instance, the rate of stricture recurrence at the ileocolonic anastomosis can be up to 70% within 12 months postoperatively) and subsequent surgical interventions predispose the patient to short bowel syndrome [11]. This apprehension induced the need for other, less radical therapeutic resources such as endoscopic therapy with the development of balloon dilation, which preserves the intestinal length and delays the need for surgery for up to 5.4 years (in the case of primary CD strictures) or over 6.4 years (in anastomotic CD strictures) [12,13]. However, the need for salvage surgery (subsequent surgery after primary EBD) is as high as 44.4% [13].

It is, therefore, critical to analyze and attain a clear understanding of the factors responsible for the unacceptably high rate of surgical interventions despite the therapeutic advances that have been achieved in the last two decades. Thus, we aimed to assess the factors associated with the risk of surgery in IBD patients. Moreover, we compared the different characteristics of the patients and their disease, depending on the presence of biologic therapy before a surgical intervention.

## 2. Materials and Methods

### 2.1. Study Design and Setting

This was a multicenter retrospective cohort study of patients with IBD from three tertiary care hospitals in Bucharest, Romania (Fundeni Clinical Institute, Emergency Clinic Hospital and St. John Emergency Hospital). All data were retrospectively collected from the medical records of the patients (hospitalized between January 2017 and June 2021) and were kept anonymous and in accordance with the Declaration of Helsinki.

Ethical approval (registration number 28830/May 2022) was obtained in accordance with the Health Minister Order, 1502/2016.

### 2.2. Patients

The inclusion criteria for the study enrolment were patients older than 18 years with a diagnosis of either Crohn’s disease or ulcerative colitis who underwent a surgical intervention related to the IBD.

### 2.3. Variables

More than ten variables were analyzed in our database, which was solely created for this study:Age at the time of the IBD diagnosis;Age at the time of surgery/time interval between the diagnosis and the surgical intervention;IBD phenotype according to the Montreal classification [14];Smoking status;Extraintestinal manifestations;Surgery indications;Type of intervention;Postoperative complications (early and long-term);Medical therapy (before and after surgery);Timing of biologic therapy initiation before or after surgery;Postoperative remission or recurrence.

### 2.4. Study Size

From a total of 540 IBD identified cases, 56 patients who underwent surgical interventions in the aforementioned period were ultimately included. ECCO (European Crohn’s and Colitis Organisation), BSG (The British Society of Gastroenterology) and ACG (American College of Gastroenterology) guidelines were followed in terms of the IBD management.

### 2.5. Statistical Analysis

For the statistical analysis, we used the SPSS 20.0 v.20 (IBM, Armonk, NY, USA) software package and considered a *p*-value below 0.05 to be statistically significant. The continuous variables were expressed as the means ± standard deviations and the ranges as medians and min–max ranges. The categorical variables were expressed as frequencies/absolute numbers with percentages. We performed several tests according to our databases such as chi-squared or Fisher’s test depending on the sample size; we also used the ANOVA unifactorial test and McNemar–Bowker’s test.

## 3. Results

### 3.1. Demographic Characteristics

Noticeably, 10.37% (56/540) of the previously diagnosed patients with inflammatory bowel disease required at least one surgical intervention during their disease course.

Most of the patients (45 patients, 80.35%) had surgical interventions due to CD and only 11 patients (19.64%) had interventions that were related to UC.

The male-to-female ratio was almost equal, with a slight (but negligible) male preponderance (51.79%). There were 27 women and 29 men.

The median age at the time of IBD diagnosis was 32.66 years, with a broad age range interval of diagnosis (the minimum age was 18 and the maximum age was 83 years old; the interquartile range was 18). Most patients (66%) had been diagnosed during their second and fourth decade of life (A2, according to the Montreal Classification). Only two patients had been diagnosed above their sixties.

Regarding smoking, only 6 out of the 56 patients enrolled were active smokers; all of them had CD.

### 3.2. IBD Phenotype

With regard to the disease phenotype, almost half of the surgical CD patients (22/45 patients, 48.8%) had an ileocolonic involvement (L3 Montreal), followed by colonic (16 patients, 35.5%) and ileal phenotypes (7 patients, 15.5%). The most common behavior of CD was the stricturing subtype (55%), followed by a penetrating disease (45%); nearly one-third of the latter subgroup had an associated perianal disease.

As expected, the majority of patients who underwent surgery in the UC subgroup had an extensive (pancolitis/E3 Montreal) involvement (7/11 patients, 63.6%). Only one patient had a limited disease (proctitis).

### 3.3. Extraintestinal Manifestations

In our cohort, 11 patients (19.64%) had associated extraintestinal manifestations (EIMs); these were more common in the female patients. None of them had more than one extraintestinal manifestation. With the exception of one patient with UC who had rheumatoid arthritis as a comorbidity, the remainder of the EIMs were related to CD. Of these, the most common EIMs were ankylosing spondylitis (3/11), followed by erythema nodosum (2 cases), oral aphthous ulcers (2 cases) and episcleritis (1 case). Psoriasis and osteoporosis were each noted in two patients.

### 3.4. Timing of Surgery, Type of Intervention and Indications for Surgery

The mean age for the surgical interventions was 35.19 ± 13.399 years. With a mean age of an IBD diagnosis of 32.66 ± 13.353 years, there was an interval of less than 3 years between the diagnosis and the timing of the first surgical intervention. Furthermore, half of the surgical procedures occurred in the first year of diagnosis.

Another important finding in our study was that among the surgical interventions, more than half (51.8%) were performed in an urgent setting whereas 48.2% were elective.

The most common indication for surgery was an intestinal obstruction (44%); stenoses of the terminal ileum and ileocecal valve were the most frequent obstruction sites (64%), followed by sigmoid colon (16%), ileocolonic (12%) and jejunum (4%). Multiple stenosis sites were noted only in 1 patient (4%).

Perforation, fulminant colitis and fistula/abscess formations were the second most common complications that required a prompt surgical intervention, accounting for a total of 40%.

All perforations (8/8) were recorded in CD, in both stricturing (B2) and penetrating (B3) phenotypes. Two cases out of eight perforations were iatrogenic; one perforation occurred during a follow-up colonoscopy and one case after the therapeutic balloon dilation of a transverse colon stricture in a patient known with ileocolonic CD. In both cases, a segmental colectomy with a stoma formation was performed and the postsurgical outcome was favorable.

Fulminant colitis was reported in eight UC patients. It is important to mention that most patients (6/8) had already initiated an anti-TNF biologic therapy before the complication occurred. The surgical management implied either a subtotal colectomy or a proctocolectomy with an ileostomy.

### 3.5. Postoperative Complications

Early postoperative complications (<3 months) occurred in 19.6% patients (11/56). Anastomotic leaks (4/11) and abdominal sepsis (2/11) were the most frequent complications that were not influenced by corticosteroids or biologic therapy. Other early complications were an enterocutaneous fistula, a pulmonary embolism, deep venous thrombosis (DVT), a perianastomotic abscess and a Clostridium difficile infection (1 case of each complication mentioned). Noticeably, all early postoperative complications (11/11) were related to the surgical interventions in the emergency setting.

A high rate (29%) of long-term postoperative complications (>3 months) was observed. Notably, almost half of these (7/16) were related to an ostomy due to a stoma reversal failure (3/7, owing to the disease activity), stomal stenosis (2/7, where endoscopic interventions were required), wound infection (1/7) and a peristomal abscess (1/7). Other long-term complications were anastomotic leaks (2 cases), pouch-related complications (2 cases; pouchitis and a pouch–vaginal fistula), enterocutaneous fistulas (2 cases), proctitis (2 cases; inflammation in the remnant rectum after a subtotal colectomy for fulminant colitis was observed at subsequent follow-ups and a proctectomy was performed) and an ischiorectal abscess (1 case).

The mortality rate was 1.8%. One death was reported due to generalized peritonitis secondary to an inflammatory mass perforation in the terminal ileum of a 62-year-old female with known CD.

### 3.6. Biologics before and after Surgery

There were 16 patients (28.57%) out of 56 who had biologic therapy initiated before surgery; 10 patients had infliximab, 4 patients had adalimumab, 1 patient had vedolizumab and 1 patient had ustekinumab.

Significant changes in the medical therapy were observed at the postsurgical intervention (*p* < 0.001, Table 1). An additional 30 patients started biologic therapy after surgery; 15 patients with infliximab, 11 patients with adalimumab, 2 patients with vedolizumab and 2 patients with ustekinumab.

In relation to biologic therapy, we analyzed the association between several factors (gender, age, age at surgery, time of biologic therapy initiation, IBD type and phenotype, EIM, type of surgery and early or long-term postoperative complications) and surgical interventions by dividing the patients into three categories: patients with biologic therapy initiated before surgery vs. after surgery vs. patients without biologic therapy (Table 1).

We observed that patients with biologic therapy before surgery had a lower mean age at the time of IBD diagnosis (*p* = 0.005) and also at the time of surgery (*p* = 0.005). Moreover, we identified a significantly higher number of patients with CD and no biologics before surgery who required an intervention; the opposite was observed in UC, where most patients were refractory to biologic therapy and underwent surgery (*p* = 0.016). Surprisingly, there were no statistically significant differences between the three groups in terms of the IBD phenotype, extraintestinal manifestations, type of surgery and postoperative complications.

We also compared the aforementioned variables between the CD and UC patients (Table 2). The statistically significant finding (*p* < 0.001) was that the IBD phenotype with an ileocolonic involvement in CD (L3) and extensive UC (E3) was associated with a high risk of surgery in both categories. The time of the biologic initiation, the type of surgery (elective/emergency) and the postoperative complications did not seem to influence the rate of surgical interventions between the two groups.

## 4. Discussion

Noticeably, the rate of surgical interventions in our study was lower than previously mentioned in other retrospective studies (10.3%). As observed by Bernstein et al., the cumulative surgery rates were 10–35%, 21–59% and 37–61% at 1, 5 and 10 years after diagnosis, respectively [15]. A trend, however, for lower surgical rates in Eastern Europe was noted (21.3% for 5 years after the diagnosis) [16]. More than half (66%) of our patients were diagnosed between 20 and 40 years. This observation could confirm the results of prospective studies showing that a young age at diagnosis is a poor predicting factor for IBD evolution and surgery interventions [17].

There are a few possible observations that might explain the lower surgery rate in our study. First, it is important to mention that the timeframe of the observations was shorter in our cohort (60 months) compared with the previously mentioned studies. Second, there was a good response to infliximab for fistula healing (>70% patients obtained a fistula closure in a median time of 12 weeks). Third, in around one-fifth of cases, a top-down medical approach was favored, with the early initiation of biologic therapy in patients with extensive disease suggestive of a potentially aggressive disease course. There are data that indicate that the early introduction of biologic therapy may improve the disease outcome through a faster remission, reducing the use of corticosteroids and reducing or delaying the need for surgery. It also has a positive impact regarding financial costs in luminal moderate-to-severe disease [18,19,20]. According to King et al., the colectomy rates during a ten-year period (2007–2017) dropped in the United Kingdom by 15% for acute UC admissions synchronously with a 4x increase in the use of biologic therapy [21].

Nevertheless, a high rate of surgical interventions in the first year of diagnosis was observed; half of the surgical interventions were performed in this time interval. In view of this concern, patient data were analyzed; a young age of diagnosis, ileal involvement and signs of stenosis were the main factors that seemed to contribute to the fulminant disease evolution. These data differ quite significantly compared with other European studies; for instance, Chaparro et al. recently published a study where only 6% of the patients underwent surgery in the first year of diagnosis [22]. Therefore, several potential explanations have emerged. One explanation might be attributed to the delayed presentation of patients, who might postpone seeking medical advice. As observed by Zaharie et al., a long diagnostic delay in CD correlated with bowel stenoses (OR 3.38) and IBD-related surgery (OR 1.95) [23]. Similar results were noted by Schoepher et al., with the advantage that his study also included regional hospitals as well as university centers [24]. Second, doctors might defer a proper diagnosis due to the misattribution of symptoms to other, more common functional gastrointestinal disorders such as irritable bowel syndrome especially in the young, where there might be a constraint to recommend thorough investigations and invasive procedures such as a colonoscopy. As Halpin et al. noted, nearly 50% patients with IBD have symptoms that meet the symptom-based criteria for the diagnosis of IBS [25]. Another potential explanation could be the need for a more cautious recognition of potentially aggressive disease courses where a top-down treatment approach (starting directly with biologic therapy) might be beneficious. The TOP-DOWN trial demonstrated a net beneficial effect of this strategy; 60% patients in the early immunosuppression group were surgery-free compared with 35.9% of those with a conventional management [18].

In terms of the disease phenotype, almost half of surgical CD patients had an ileocolonic involvement and 55% had an associated stricturing disease. These data were similar to those found in a Danish population study group, where Lo et al. also found that ileal involvement and the stricturing subtype were associated with a high risk of surgery [26]. In UC patients, more than 60% had an extensive involvement (pancolitis). Comparably, Manser et al. found an OR of 2.5 for surgery in UC patients with extensive disease from the Swiss IBD Cohort Study [27].

A high rate of postoperative complications, primarily ostomy-related, was observed in our study, which could be explained by the fact that almost half of the surgical interventions (48%) were performed in an urgent setting (<48–72 h from the index emergency department presentation). Compared with other studies, this rate of urgent setting surgery was significantly higher than the other evidence we found. Lowe et al. queried more than 1,795,000 IBD-related hospitalizations during an 8-year period and found a stable proportion of urgent surgeries for CD (25%) and a decreased trend for UC (from 21 to 14%) [28].

Regarding extraintestinal manifestations, we found similar data to Rogler et al., who mentioned a 24% incidence of EIMs; this could be clinically relevant even before the index manifestation of IBD [29]. Vavricka and his colleagues found in the Swiss IBD Cohort that around one-quarter of patients may have had an EIM before the IBD diagnosis; in 74% of cases, the first EIM manifested after the IBD diagnosis [30].

Concerning medical therapy, there was a high rate of postoperative switching therapy (5/11 patients) due to a lack of response, low drug therapeutic levels or the high titer of anti-drug antibodies. Consequently, optimizing biologic therapy in IBD is critical, with a necessity for active and personalized therapeutic drug monitoring (for instance, a higher drug level might be needed for perianal fistula healing and a lower trough level to prevent CD recurrence postoperatively). The results of a multicenter study on 264 IBD patients demonstrated that proactive drug monitoring (DM), in comparison to reactive DM, was associated with a reduced risk of treatment failure, IBD-related surgery/hospitalization and a reduced rate of antibodies to infliximab [31].

In our retrospective study, only 28.57% patients had biologic therapy initiated before surgery compared with 53.5% patients after surgery, which brought into question whether an earlier biologic therapy would have been beneficial to patients to an even greater extent. Both pre-and postoperatively, infliximab and adalimumab were the most frequent biologic therapies used. According to AGA (American Gastroenterological Association) guidelines, the early initiation of biologic therapy within 8 weeks after surgery is considered to be safe and beneficial [32]. The present study showed that biologic therapy did not have a significant statistical impact on the type of surgery (emergency vs. elective) and had no influence in terms of early or long-term postoperative complications. There are other studies that confirm the safety of biologic therapy used perioperatively; for example, Gainsbury et al. compared 29 UC patients treated with infliximab within 12 weeks of IPAA with 52 control patients who underwent IPAA without a recent infliximab infusion, with no significant differences in terms of operative mortality or postoperative complications [33].

The limitations of the study were related to the lack of a control group. Other limitations were the low number of the sample and the retrospective nature of the study.

## 5. Conclusions

To our knowledge, this is the first report in Romania to evaluate IBD patients that underwent surgical interventions and the impact of biologic therapy regarding the outcomes and complications. In our group, we observed that patients who underwent surgery were young at the diagnosis, had ileocolonic disease (CD) and extensive disease involvement (UC). Patients with biologic therapy initiated before surgery were younger. Most surgical interventions were observed in the first year of diagnosis. A high number of patients with CD and no biologics before surgery required an intervention; the opposite was noted for UC. We found no influence of biologic therapy in terms of the IBD phenotype, extraintestinal manifestations or type of surgery (emergency/elective). Postoperative complications also seemed to be related to the emergency surgical setting. Biologic therapy had no significant statistical impact.

In conclusion, the management of patients with inflammatory bowel disease has become increasingly complex and challenging, requiring a multidisciplinary approach. The unpredictable pattern evolution of the disease and the struggling encountered by clinicians to achieve long-term remission has fueled the development of novel molecules and it is critical to acknowledge the poor prognostic factors leading to a potentially aggressive disease behavior and last but not least, the importance of the proper indications and timing of biologic therapy and surgery in order to ensure the best possible care for patients.

## Figures and Tables

**Table 1 medicina-59-00337-t001:** Comparison between patients with biologics initiated before surgery vs. patients with biologics initiated after surgery vs. patients without biologics.

Variables	Biologics Initiated before Surgery (*n* = 16)	Biologics Initiated after Surgery (*n* = 30)	No Biologics (*n* = 10)	*p*-Value
Gender (M/F (%))	7/9 (43.8/56.2%)	19/11 (63.3/36.7%)	3/7 (30/70%)	0.153
Age, mean ± SD	27.13 ± 8.85	31.87 ± 10.71	43.90 ± 19.79	0.005
Age at surgery, mean ± SD	29.38 ± 10.84	34.60 ± 10.42	43.30 ± 18.77	0.005
Smoking (yes/no)	1/15 (6.2/93.8%)	5/25 (16.7/83.3%)	0/10 (0/100%)	0.376
Time of biologic initiation				0.217
<3 months	5 (31.2%)	16 (53.3%)		
>3 months	11 (68.8%)	14 (46.7%)		
IBD type				0.024
CD	9 (43.8%)	27 (90%)	9 (90%)	
UC	7 (56.2%)	3 (10%)	1 (10%)	
IBD phenotype				0.221
L1/L2/L3	1/2/6 (6.25/12.5/37.5%)	5/12/10 (16.7/40/33.3%)	1/2/6 (10/20/60%)	
E1/E2/E3	1/2/4 (6.25/12.5/25%)	0/1/2 (0/3.3/6.7%)	0/0/1 (0/0/10%)	
Extraintestinal manifestations (yes/no)	3/13 (18.8/81.2%)	8/22 (26.7/73.3%)	0/10 (0/100%)	0.208
Type of surgery (elective/emergency)	7/9 (43.8/56.2%)	13/17 (43.3/56.7%)	7/3 (70/30%)	0.346
Early postoperative complications (yes/no)	4/12 (25/75%)	6/24 (20/80%)	1/9 (10/90%)	0.671
Long-term postoperative complications (yes/no)	6/10 (37.5/62.5%)	9/21 (30/70%)	1/9 (10/90%)	0.375

**Table 2 medicina-59-00337-t002:** Comparison of CD and UC subgroups.

Variables	CD (*n* = 45)	UC (*n* = 11)	*p*-Value
Gender (M/F (%))	23/21 (51.1/48.9%)	6/5 (54.5/45.5%)	0.838
Age, mean ± SD	32.44 ± 14.41	33.55 ± 8.11	0.809
Age at surgery, mean ± SD	35.09 ± 14.48	35.64 ± 8.04	0.905
Smoking (yes/no)	6/39 (13.3/86.7%)	0/11 (0/100%)	0.334
Biologics (yes/no)	36/9 (80/20%)	10/1 (90.9/9.1%)	0.667
Time of biologic initiation (<3 months/>3 months)	15/21 (41.7/58.3%)	6/4 (60/40%)	0.475
IBD phenotype			<0.001
L1/L2/L3	7/16/22 (15.55/35.56/48.89%)	0/0/0 (0/0/0%)	
E1/E2/E3	0/0/0 (0/0/0%)	1/3/7 (9.1/27.3/63.6%)	
Extraintestinal manifestations (yes/no)	10/35 (22.2/78.8%)	1/10 (9.1/90.9%)	0.434
Type of surgery (elective/emergency)	22/23 (48.9/51.1%)	5/6 (45.5/54.5%)	0.883
Early postoperative complications (yes/no)	9/36 (20/80%)	2/9 (18.2/81.8%)	0.631
Long-term postoperative complications (yes/no)	12/33 (26.7/73.3%)	4/7 (36.4/63.6%)	0.711

## Data Availability

Additional data are available upon request from the first author.

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
