# Peer review of "Surgery for Inflammatory Bowel Disease in the Era of Biologic Therapy: A Multicenter Experience from Romania"

_medicina, 2023, doi:10.3390/medicina59020337_

Round 1
Reviewer 1 Report (New Reviewer)
Manuscript provides a retrospective review of data from three Romanian hospitals that focuses on biologic usage and type of IBD. While the concept of the manuscript is interesting and has a focused population receiving at least one surgery, some of the conclusions do not reflect a control group of not having surgery. Some changes to the wording on some of the conclusions should be changed. Considerations should be given to the statistical methods in regards using a Chi-square test vs. Fisher’s exact test, and additionally to how values our presented.
In section 2.3, the ranges mention. Are they min-max ranges or interquartile range? Need to specify.
Line 120 51,78% should be 51.75%
The same question/comment remains for all of the reported ranges throughout the results section as mentioned above. Based on the inclusion criteria it looks like the range in line 122 is a min/max. What is the interquartile range?
Line 124-125s conclusion regarding age is a very strong statement without context of the populations seen at the hospitals of interest. Also need to consider other confounding factors or competing factors that would not allow individuals to reach the age of 60+
In lines 147 and 148, provide SD of age at surgical interventions and IBD diagnosis.
Is mortality rate of 1.8% on the cohort of 56 or of those that had post operative complications
For the categorical variable in Table 1 & 2, chi-sq test are not appropriate. Should use Fisher’s exact since there are cells less than 5.
Table 1- what is the p-value comparing for Gender? Is it gender in total by the 3 groups. Should be Fisher’s Exact Test.
What is the comparison for IBD phenotypes (in both Table 1 &2)? The single p-value here does not make sense. The division of the cells in Table 2 appear to be incomplete.
“et al” in line 224 has a different format
What considerations were given to elective surgeries during a the COVID19 pandemic? What considerations were given for site differences and approaches to care? What are the limitations of this disease within Romania (i.e., prevalence, incidence, etc.)?
Line 307-308 “High risk factors for surgery were young age at diagnosis, ileo-colonic 307 disease (CD) and extensive disease involvement (UC)” does not have supporting evidence in this manuscript. The results presented are structured around time of biologics and type of IBD (UC vs. CD). There are no analyses that look at those that had surgery vs. those that did not which is what line 307 and 308 indicate. All analysis are subset to those that solely had at least one surgical intervention. What about those that didn’t have surgery? How do the variables you’ve collected show up in that sample.
Author Response
Please see the attachment.

Reviewer 2 Report (New Reviewer)
The principal idea is really interesting, the comparison of the effect of preoperative biological treatment with no biological treatment is relevant, the comparison of postoperative biological treatment regarding emergency surgery and preoperative complications is however not logical. It is not clear to what extent preoperative infliximab treatment could play a role in the completion of fibrotic stricture, that is not unfrequently occuring following the introduction of infliximab treatment in young CD patients with ileo-coecal involvement. If so, it would be interesting, how long was the delay to the surgery in those patients. On the other hand more detailed data about immunosupressive treatment in the different patient-groups would be desirable.
Author Response
Please see the attachment.

Reviewer 3 Report (New Reviewer)
The authors mainly assessed the risk factors for surgery in IBD patients undergoing biological therapies. I have several questions and concerns for the authors:
ABSTRACT
- Please state the number of patients included. Also, please mention all potential risk factors you evaluated in this study.
INTRODUCTION
- Please cite and address the points reported by recent studies relevant to your topic (https://doi.org/10.1007/s11605-020-04563-0 ; https://doi.org/10.1093/gastro/goz004)
METHODS
- Please report your study as per the STROBE guideline in more detail.
- Please add a reference for the IBD phenotype based on the Montreal classification.
- In the statistical analysis section, please write which tests were used for what variables.
DISCUSSION
- It would help if you discussed your findings on the factors assessed about the surgery in more detail.
- What were your limitations?
Round 2
Reviewer 1 Report (New Reviewer)
The edits that have been added to the manuscript enhance its overall merit and understanding. The limitations with this retrospective study design are typical and results are still beneficial to the scientific community. Thank you for cleaning up the tables and clarifying your statistics.
Reviewer 3 Report (New Reviewer)
Thank you for your responsive revisions.
This manuscript is a resubmission of an earlier submission. The following is a list of the peer review reports and author responses from that submission.
Round 1
Reviewer 1 Report
The article “Surgery for inflammatory bowel disease in the era of biologic therapy- A multicenter experience” by Christopher Pavel et all, is a paper focused on the potential role of biologics in modifying the surgical history of IBD patients. Authors retrospectively collected data from 3 tertiary care hospitals of Bucharest (Romania) between January 2017 and June 2021. They identified 56 IBD patients who underwent surgery in the aforementioned period. 16 out 56 were on biologics at time of surgery. Authors reported no statistically significant differences between patients receiving biologics before surgery or not, in terms of type of surgery, postoperative complications and IBD phenotype.
Unfortunately the paper doesn't show any real novelty and the sample size is too small to reach definitive conclusions. Even more the study considered both CD and UC patients that are scarcely comparable in a surgical setting due to the real different type of surgery.
Reviewer 2 Report
According to the guidelines for authors, write the title in lowercase letters.
In the title, it should be pointed out that this is a multicenter experience from Romania
It is not necessary to include section subheadings within the abstract.
In the summary, instead of the abbreviation CD, write the full name of the word.
I also ask for the full name and for all abbreviations within the text that appear for the first time.
Please provide ethical approvals from all three hospitals from which the data were collected.
The first sentence in the statistical analysis section is unnecessary.
In your research, you used both the mean +/- standard deviation and the median (interquartile range). Have you performed a test for normality of the distribution? How did you decide which test to use?
Figures 1 and 2 are unnecessary since they are described in the text. Please remove them.
Please pay attention to the rounding of percentages...eg. 51.78 + 48.21 = 99.99
Can fistula be considered an immediate complication? If yes, clearly state that these are complications within 3 months.
On the other hand, can we consider anastomosis leakage and abscess as late complications?
Both Figure 4 and Figure 5 are also described in the text so they are not required.
It is highly ungrateful to evaluate and compare outcomes based only on whether biological therapy was initiated before or after surgical intervention. It would be necessary to know exactly how long it was from the beginning of the application of biological therapy to the actual surgical intervention. With this logic, you would include in one group a patient who started biological therapy the day before surgery, while in the other you would include a patient who started biological therapy the day after surgery. What conclusions can be drawn from that?
It is clear in the discussion that other studies have a much larger sample of patients from which stronger evidence and conclusions can be drawn.
The conclusion is formulated too generally and universally.